# Consumption of animal source food and associated factors among pregnant women in eastern Ethiopia: A community-based study

Meseret Belete Fite[1]*, Abera Kenay Tura[2,3], Tesfaye Assebe Yadeta[2], Lemessa Oljira[4], Kedir Teji Roba[2]

**1** Department of Public Health, Institute of Health Sciences, Wollega University, Nekemte, Ethiopia, **2** School of Nursing and Midwifery, College of Health and Medical Sciences, Haramaya University, Harar, Ethiopia, **3** Department of Obstetrics and Gynaecology, University Medical Centre Groningen, University of Groningen, The Netherlands, **4** School of Public Health, College of Health and Medical Sciences, Haramaya University, Harar, Ethiopia

* meseretphd2014@gmail.com

## Abstract

### Introduction

Animal source foods contain quality nutrients, immunity, and behavioral outcome and are important for growth, and development. However, evidence on the level of animal source food consumption frequency and associated factors among pregnant women in Ethiopia, particularly rural residents are limited. Therefore, this study aimed to assess the consumption frequency of animal source food and to identify associated factors among pregnant women in the Haramaya district.

### Methods

A community-based cross-sectional study was conducted among 448 pregnant women. Data were collected through face-to-face interviews by trained research assistants, using a validated frequency questionnaire. Consumption of animal food sources was assessed by counting the frequency of each food from animal sources that pregnant women ate over a seven-day reference period. The highest tertile for animal source food consumption was considered as the high frequency of animal source food consumption; whereas the two lower tertiles were taken as the low frequency of animal source food consumption. A binary logistic regression model was used to investigate the association of the independent variables with the animal source food consumption. An adjusted odds ratio with a 95% confidence interval was reported to show an association using a p-value <0.05.

### Results

The high frequency of animal source food consumption among the study participants was 24.78% (95% CI = 21%-29%). High animal source food consumption was more likely higher among respondents who were literate (AOR = 1.80; 95% CI = 1.048–3.095), and those who owned milk cows (ARO = 1.70; 95% CI = 1.003–2.863). However, respondent who reported

**

**Data Availability Statement:** All relevant data are within the paper.

**Funding:** This study was fully funded by Haramaya University, Ethiopia. The funder has no role in the

conception; design of the study, statistical analysis, and result interpretation, and in writing up the manuscript. The funding institution has no role in the publication consent or approval.

**Competing interests:** The authors have declared that no competing interests exist.

chewing khat (AOR = 0.51; 95% CI = 0.313–0.805) (AOR = 0.56; 95% CI = 0.349–0.903), were less likely experienced animal source food consumption.

## Conclusion

We found low animal source food consumption among pregnant women in this predominantly rural setting. Women's educational level and milk cow ownership were positively associated with animal source food consumption. Additionally, a lower frequency of animal source food consumption was observed among women who reported chewing khat. Therefore, nutrition policy programs and interventions aimed at encouraging maternal nutritional guidance and counseling are recommended.

## Introduction

Malnutrition is an important public health issue. Worldwide, among 676 million under-five children, about 155 and 52 million are stunted wasted respectively [1–3]. Malnutrition can be displayed as to whether growth failure or micronutrient deficiency and aggregation of predictors bring about malnourishment [4]. For instance, low consumption of animal source food (ASF) has been documented to augment, the danger of malnutrition [5–7]. ASFs are a better source of quality protein and essential micronutrients of special importance for pregnancy outcomes, the health, and the development of infants [8–10]. Furthermore, the introduction of a slight amount of ASF in a diet can enhance the nutritional enactment of plant-based foods and consumers' nutritional status [11,12]. Nevertheless, there are yet many communities globally that have low or marginal access to ASF [13]. Numerous food consumed by developing countries are lacking an essential quality and the quantities of energy, protein, and other nutrients to suggestions [14]. Having access to animal source foods contributes an important function in a well-balanced diet by supplying nutrients that are essential to life and needed for healthy growth, development, and functioning [15].

High consumption of ASFs is observed to be significantly associated with pregnancy outcomes and birth outcomes such as improved growth, cognitive function, physical activity levels, school performance, and morbidity in young children [16,17]. Therefore, the intake of ASFs can encourage dietary diversity and nutrition in pregnant women [18]. Inadequate dietary consumption in pregnancy is an important contributor to global maternal malnutrition in less developed countries [19]. A previous study showed that pregnant women in developing countries suffer from energy deficiencies due to comparatively inadequate energy intake [20]. Cultural norms and customs govern dietary intake behaviors in several traditional societies comprising critical life stages such as pregnancy [21]. Meat and egg are taboo among pregnant women in South Eastern Nigeria [22]. Ethiopia is noted to have one of the substantial livestock populations globally [23]. Nevertheless, a low intake of meat, fish, fruits, and some vegetables during pregnancy is reported [24]. In predominantly rural settings, ASF is commonly taken during extra special family/ public events as it is contemplated as an enjoyment ·diet instead of a crucial portion of the regular family diet [25,26].

We hypothesized that pregnant women in this study setup exhibit a lower frequency of animal source food consumption, and this is affected by different independent predictors. Therefore, the objective of this study is to assess the level of animal source food consumption and associated factors among pregnant women.

## Methods

### Study settings

The study was embedded into the Haramaya Health Demographic Surveillance and Health Research Centre (HDS-HRC), which was established in 2018. The HDS-HRC is located in the Haramaya district. Haramaya District is located 500 km away from the capital city, Addis Ababa to the east. Haramaya district consists of 33 kebeles (the lowest administrative unit in Ethiopia). HDS-HRC covers 12 rural kebeles which are representative and randomly selected by considering geographic and environmental issues. In HDS-HRC 2306 pregnant women were followed. The district has mixed farming, with the major cash crop being khat (Catha edulis Forsk) [27]. The study was conducted from January 5 to February 12, 2021.

### Study design and population

A community-based cross-sectional study was conducted. All pregnant women living in the district constituted the source population; whereas all pregnant women who lived in the selected kebeles for at least six months during the study period were the study population. Whereas, those who were critically ill during data collection were excluded from this study. The sample size was determined using single and double population proportion formulas with their corresponding assumption and the largest sample was considered. As such, the sample computed using single population proportion formula with the following assumptions gave the largest sample (n = 393): 95% confidence interval, level of the high frequency of consumption of ASF among pregnant women in West Gojjam Zone, Northwest Ethiopia (36.6%) [28], 5% marginal error and 10% non-response rate. However, this study is a part of a larger longitudinal study that obtained birth outcome information from pregnant women. Thus, the sample size used in this study was calculated from the larger study that included 475 pregnant women. After constructing the sampling frame from the HDS-HRC database, simple random sampling was applied to randomly select eight kebeles and then eligible women using a computer-generated lottery method.

### Data collection and measurement

Data were collected through interview administered questionnaires by trained research assistants. The questionnaire contained data on socio-economic, obstetric, maternal perception, food consumption, dietary knowledge, attitude, and practices of pregnant women. Structured questionnaires that are adapted from the review of literature were initially prepared in the English language and were translated to the local language (Afan Oromo) by an individual with good command of both languages. It was also pre-tested on 10% of the sample in Kersa District before data collection. In addition, mid-upper arm circumference (MUAC) was measured to assess nutritional status.

The validated food frequency questionnaire (FFQ) containing 27 of the most common lists of food items consumed by the district community was used to assess the dietary practices of the study participants [29–32]. Additionally, this validated FFQ was used to assess the dietary diversity of the participants [33,34]. Initially, the list of food items was established based on consultation of key informants living in the study area, who knew the culture, local language, and foods typically consumed. Then the food frequency questionnaire was pretested on 10% of the sampled pregnant women in the district who were not included in the main study and necessary modifications were made based on the observations. In addition, pretested food frequency questionnaires were carried out on 10% of the sampled pregnant women of the district not included in the main study. Necessary modifications were made before actual

implementation to generate data. Finally, to measure the consumption of each food per day, per week, or month for the FFQ in the past three months consider the difference in dietary consumption within a day of a week to take the concept into account. However, we considered the greater difference in dietary practice in the local community over the day of the week, and the intake of each food item per day [28,35] was not taken as a cut-off point to label consumers. In doing so, pregnant women were defined as "consumers" of a food item if they had consumed those items at least once over a week [33,36].

The consumption of foods from an animal source (ASF) was estimated by counting the frequency of each food from animal sources that pregnant women ate over a reference period. Animal source foods score was also converted into tertile and the highest tertile was used to label as "highs, while the two lower tertiles combined were defined as "low" ASF. The food items in the FFQ were grouped into ten food groups. These are cereal, white roots and tubers, pulse and legumes, nuts and seeds, dark green leafy vegetables, other vitamin A-rich fruits and vegetables, meat, fish and poultry, dairy and dairy product, egg, other vegetables, and other fruits [35]. The sum of each food group that the pregnant women consumed over one week were calculated to analyze the dietary diversity score (DDS).

Furthermore, the dietary diversity score was converted into tertiles, and the highest tertile was used to label a "high" dietary diversity score whereas both lower tertiles combined were defined as a" low" dietary diversity score. The food variety score (FVS) is the frequency of individual food items consumed in the reference period of the study. Therefore, it was estimated by the intake of 27 food items by each individual over seven days [33], with a maximum of FVS fourth. Finally, the mean FVS of pregnant women was calculated and those of them with FVS greater than the means were labeled as having a "high" food variety score whereas those with FVS lower than the means were defined as having "low" FVS.

## Data quality assurance

Two days of rigorous and extensive training with the final version of the questionnaires were given to each data collector and supervisor before the pre-test. Collected data was checked by supervisors before being sent to the data entrée on daily basis. We pre-tested the questionnaires on 10% of the sampled pregnant women of the kersa district, that were not included in the main study, and modification was done based on the pre-test observations. The supervisors kept the alleyway of the field procedures and checked the completed questionnaires daily to approve the accuracy of the data collected, and the research team managed the overall work of data collection.

## Data processing and analysis

Data were double entered using EPiData version 3.1 software. Data were cleaned, coded, and checked for missing and outliers, for further analysis and exported to STATA version 14 (College Station, Texas 77845 USA) statistical software. The outcome variable was dichotomized as animal source food consumption = 1(high frequency of AFS consumption) and animal source food consumption = 0 (low frequency of AFS consumption). Bivariate analysis and multivariable analyses were done to see the association between each independent variable and outcome variables using binary logistic regression. The assumptions for binary logistic regression were cheeked. The goodness of fit was checked by Hosmer-Lemeshow statistic and omnibus tests. All variables with p<0.25 in the Bivariate analyses were included in the final model of multivariable analysis to control all possible confounders. Multico-linearity test was carried out to see the correlation between independent variables by using the standard error and collinearity statistics (variance inflation factors >10 and standard error >2 were considered suggestive of

the existence of multi co-linearity). The direction and strength of statistical association were measured by an odds ratio of 95% CI. Adjusted odds ratio along with 95%CI was estimated to identify factors associated with animal source food consumption. Correlation between independent variables was checked using the Pearson Correlation Coefficient. P-value < 0.2 was used as a cut-off point to select variables for the final model. Backward elimination was used, and P-value < 0.05 was considered statistically significant.

To estimate the economic level of the families, a wealth index was employed. The wealth dispersion was generated by applying principal component analysis. The index was calculated based on the ownership of latrine, selected household asset, quantity of livestock, and source of water used for drinking, that was 41 household variables. Nutritional knowledge of the women was gauged through 16 nutritional knowledge questions on the feature of nutrition needed in their course of pregnancy. Lastly, the highest tertile was defined as having "Good" nutritional knowledge and the two lower tertiles were labeled as "Poor" nutritional knowledge. The maternal attitude was evaluated with 12 Likert scale questions using PCA. The factor scores were totaled and classified into tertiles (three parts), and the highest tertile was defined as having a "Favorable" maternal attitude and the two lower tertiles were characterized as "Unfavorable" maternal attitude. The maternal perceived vulnerability to malnutrition was evaluated with 10 Likert scale questions using PCA. The factor scores were totaled and classified into tertiles (three parts), and the highest tertile was defined as having a perceived vulnerability "Yes" and the two lower tertiles were characterized as "No" maternal perceived vulnerability. Similarly, perceived severity of malnutrition, perceived benefit to healthy nutrition perceived barrier to healthy nutrition, and perceived self-efficacy to control malnutrition during pregnancy was calculated by using their composite questions. Women's autonomy was evaluated by seven validated questions which were adopted from the Ethiopian demographic health survey [34]. For each response to a question, the response to each question was coded as "one" when the decision was made by the pregnant woman alone or jointly with their husband, otherwise "zero.

### Ethical consideration

All methods of this study were carried out in accordance with the Declaration of Helsinki-Ethical principle for medical research involving human subjects. An ethical approval letter was obtained from Haramaya University Institutional Research Ethics and Review Committee (IRERC) with a reference number of (IHRERC/266/2020) before the commencement of data collection. Written informed consent to participate was obtained from participants and legally authorized representatives "of minors below 16 years of age and illiterates" and their privacy and confidentiality were maintained. All personal identifiers were excluded, and data was kept confidential and used for the proposed study only.

## Results

### Socio-demographic characteristics

A total of 475 pregnant women were eligible, and 448 consented, making a response rate of 94.3%. The mean age of the women was 25.68 (+5.1), ranging from 16 to 36. The majority of the respondents could not read or write (73.88%), were housewives (96.1%), farmers (93%), and had a family size of 1–5 (76.56%). Only 20.09% of the respondents were in the richest quintile. Of the respondents, about 60.49% of households owned different amounts of agricultural land. Concerning domestic animal ownership, about 50.22% of households owned goats, and 26.12% owned cows, Table 1.

**Table 1. Socio-demographic of pregnant women in Haramaya District, eastern Ethiopia, 2021 (n = 448).**

| Variables | Frequency(n) | Percentage (%) |
|---|---|---|
| Age (years) | | |
| <18 | 25 | 5.58 |
| 18–35 | 400 | 89.29 |
| >35 | 23 | 5.13 |
| Mean (± SD) | 25.68 (± 5.16) | |
| Educational level of the woman | | |
| Can't read or write | 331 | 73.88 |
| Read or write | 26 | 5.81 |
| Formal education | 91 | 20.31 |
| Educational level of husband | | 49(23.33) |
| Can't read or write | 259 | 57.81 |
| Read or write | 61 | 13.62 |
| Grade 1–8 | 102 | 22.77 |
| Grade 9 and above | 26 | 5.8 |
| Occupation of the woman | | |
| Housewives | 433 | 96.65 |
| Merchants | 15 | 3.65 |
| Occupation of husband | | |
| Farmers | 420 | 93.75 |
| Daily labors | 28 | 6.25 |
| Family size | | |
| 1–5 | 343 | 76.56 |
| ≥5 | 105 | 23.44 |
| Agricultural land possession | | |
| No | 271 | 60.49 |
| Yes | 177 | 39.51 |
| Domestic animal ownership | | |
| Ox | 10 | 10.71 |
| Cow | 112 | 26.12 |
| Goat | 225 | 50.22 |
| Sheep | 79 | 17.63 |
| Wealth Index (Quintile) | | |
| Poorest | 90 | 20.09 |
| Poor | 90 | 20.09 |
| Middle | 89 | 19.87 |
| Rich | 90 | 20.09 |
| Richest | 89 | 19.87 |

## Consumption of animal source foods

Poultry products were not consumed in 97% of pregnant women over seven days before the survey. Meat (sheep/lamb, goat, beef/cattle, and any other animals) was consumed one time per week by 7.59% of the respondents. Fish products were only consumed once and more times per week by 1.34% of the respondents. Eggs were consumed once and more times per week by 4.08% and one time per week by 10.71%. However, milk and milk products were consumed once and more times per week by 4.91% and one time per week by 11.39%. The prevalence of the high frequency of animal source food consumption among the study participants

**Table 2. Consumption of ASF by pregnant women in Haramaya District, eastern Ethiopia, 2021 (n = 448).**

| Variables | Consumption frequency | Number (n) | Percentage (%) |
|---|---|---|---|
| Poultry | One and more times per week | 13 | 2.90 |
| | Never consumed | 435 | 97.10 |
| Meat (sheep/lamb, goat, beef/cattle and any other animals) | One and more times per week | 3 | 0.67 |
| | On times per week | 34 | 7.59 |
| | Never consumed | 411 | 91.74 |
| Fish products | One time per week | 6 | 1.34 |
| | Never consumed | 442 | 98.66 |
| Eggs | One and more times per week | 18 | 4.08 |
| | One time per week | 48 | 10.71 |
| | Never consumed | 382 | 85.27 |
| Milk and milk products (yogurt, cheese, etc.) | One and more times per week | 22 | 4.91 |
| | One time per week | 51 | 11.39 |
| | Never consumed | 375 | 83.70 |
| Animal Source Foods (ASFs) | | | |
| Low | | 337 | 75.22 |
| High | | 111 | 24.78 |
| Food Variety Sore (FVS) | | | |
| Low | | 280 | 62.50 |
| High | | 168 | 37.50 |
| Dietary Diversity Score (DDS) | | | |
| Low | | 316 | 70.54 |
| High | | 132 | 29.46 |
| Meal frequency | | | |
| < 4 | | 331 | 73.88 |
| ≥ 4 | | 117 | 26.12 |

was 24.78% (95% CI = 21%-29%). Of the total respondents, 29.46%, 37.50%, and 26.12% of them had high dietary diversity, high food variety score, and > 4 meal frequency respectively, Table 2.

## Factors associated with animal source food consumption

In the bi-variable analysis, women's educational level, family size, Antenatal care, perceived severity of malnutrition, milk cow ownership, sheep ownership, ox ownership, and khat chewing were found to be a candidate for multivariable analysis at p<0.25. Using logistic regression models, high animal source food consumption was more likely higher among literate respondents (AOR = 1.80; 95% CI = 1.048–3.095) and those who owned milk cows (ARO = 1.70; 95% CI = 1.003–2.863). However, respondent who reported chewing khat (AOR = 0.51; 95% CI = 0.313–0.805) (AOR = 0.56; 95% CI = 0.349–0.903), were less likely experienced animal source food consumption, Table 3.

## Discussion

The aim of the current study was twofold:(1) to assess the level of frequency of animal source food consumption and (2) to determine the factors associated with the high frequency of ASFs consumption among pregnant women. The level of the high frequency of ASFs consumption among the study participants was 24.78% (95% CI = 21%-29%) and was noted to be sub-optimal. We found that meat was not consumed in 91.74%, poultry products were not consumed

**Table 3. Factors associated with animal source food consumption among pregnant women in Eastern Ethiopia, 2021.**

| Variables | Animal source food consumption | | COR(95%CI) | AOR (95%CI) | P-value |
|---|---|---|---|---|---|
| | High frequency of ASF (n = 111) | Low frequency of ASF (n = 337) | | | |
| Educational level of women | | | | | |
| Illiterate | 74(66.67) | 283(83.98) | 1 | 1 | |
| Literate | 37(33.33) | 54(16.02) | 2.62(1.605, 4.279) | 1.80(1.048, 3.095) | 0.033* |
| Family size | | | | | |
| < = 6 | 99(97.06) | 290(92.95) | 1 | 1 | |
| >7 | 3 (2.94) | 22(7.05) | 0.39(0.117, 1.363) | 0.37(0.104, 1.321) | 0.126 |
| Milk cow ownership | | | | | |
| No | 70 (63.06) | 261(77.45) | 1 | 1 | |
| Yes | 41(36.94) | 76(22.55) | 2.01(1.267, 3.194) | 1.70(1.003, 2.863) | 0.049* |
| Sheep ownership | | | | | |
| No | 83(74.77) | 286(84.87) | 1 | 1 | |
| Yes | 28(25.23) | 51(15.13) | 1.89(1.123, 3.188) | 1.66(0.934, 2.943) | 0.084 |
| Ox ownership | | | | | |
| No | 90(81.08) | 310(91.99) | 1 | 1 | |
| Yes | 21 (8.92) | 27(8.01) | 2.68(1.446, 4.963) | 1.61(0.916, 3.579) | 0.088 |
| Perceived severity | | | | | |
| No | 83(74.77) | 277(82.20) | 1 | 1 | |
| Yes | 28(25.23) | 60(17.80) | 1.56(0.934, 2.597) | 1.23(0.689, 2.198) | 0.483 |
| Antenatal care | | | | | |
| No | 32(28.83) | 132(39.17) | 1 | 1 | |
| Yes | 79(71.17) | 205(60.83) | 1.59(0.998, 2.532) | 1.46 (0.863, 2.478) | 0.157 |
| Chat chewing | | | | | |
| No | 58(52.25) | 122(36.02) | 1 | 1 | |
| Yes | 53(47.75) | 215(63.80) | 0.52(0.336, 0.800) | 0.56(0.349, 0.903) | 0.017 * |

COR = Crude Odds Ration; AOR = Adjusted Odds Ratio, CI = Confidence Interval at 95%, CI, and P-Value were found from the multivariable Logistic regression analysis model.

* Statistically significant at p-value <0.05.

in 97%, and fish products were not consumed in 98.66% of pregnant women over seven days before the survey. Moreover, chewing chat and restriction of the intake of some foods were identified as predictors of animal source food consumption in Haramaya District.

Adequate nutrition during pregnancy is essential for maternal and child health [37]. There is mounting evidence that insufficient consumption of a balanced and quality diet during pregnancy significantly affects fetus health and development and may result in poor birth outcomes [38]. At the beginning of pregnancy, many women lack sufficient micronutrient stores to meet the increased physiological requirements [39], and they are more vulnerable to malnutrition [40]. Several epidemiological studies indicated that the low frequency of animal source food consumption contributes to maternal under-nutrition and micronutrient deficiency in resource-limited countries [41,42]. Pregnant women's diet must supply adequate nutrients for the mother, fetus, and effective lactation. Despite this reality, in this present study, the noted frequency of animal source food consumption among pregnant women was very low. This figure was much lower than studies conducted in Gojjam in northwest Ethiopia [28] and rural communities of Ethiopia [43]. The lower consumption of poultry and fish products in the present study was in line with other studies [43–45], that presented restricted inclusion of ASF in the diets of families in low-and middle-income countries (Ethiopia, India, China, and Latin

America). Nevertheless, there are findings in which improved ASF consumption figures were documented. In Ethiopia, consumption of meat was documented to range from 26.8% to 80% [46–48]. However, due to the differences in the study area and socio-cultural conditions, it is noteworthy to mention that the direct comparison of our results with previous investigations employed in Ethiopia is impossible. Another possible reason for the variation might be, as the findings were carried out in towns, the variation in dietary habits from the current study could be affected by the accessibility or availability of retailer butcher houses and households' interest in wages and the ability to purchase. The practices of regular meat intake in the prompt local culture might also have impacted. Furthermore, since this study used a one-week survey to assess the frequency of consumption of ASFs; the difference in methods and measures applied could contribute to the discrepancy.

Education is an essential instrument to equip human beings to decisively affect securing the necessities of life. In the current study, pregnant women with better educational levels were more likely to consume ASFs than those women who never attended formal education. This result is consistent with the former studies conducted in rural Ethiopia [43], Vietnam [49], and Nepal [50]. This might be because literate women have a better understanding of the importance of consuming a quality diet in pregnancy, and they may positively influence them to have high-frequency consumption of ASFs. Pregnant women who received dietary guidance are expected to consume ASFs and intake a diversified diet compared to those who do not receive nutritional advice.

The frequency of consumption of ASF is supposed to rise when respondents own domestic animals as a source of food commodities and income for diversified diets. From the findings of the current study, households owning cows were more likely to consume ASF than households not owning cows. This result is comparably in agreement with a study conducted in rural Ethiopia [43]. Availability and relative affordability may contribute to the more frequent consumption of ASF.

Even though chewing khat is an especially disseminating act in Ethiopia and developed countries, comprising Africa and Europe [51–53], health consequences are well understood. The result of the current study highlights the importance of increasing the frequency of consumption of ASFs during pregnancy with proper interventions. Therefore, pregnant women should frequently be advised of the negative consequences of chat chewing and supported to improve their dietary consumption in pregnancy.

The strengths of this study include the following: validated food frequency questionnaires were used to assess the frequency of consumption of ASFs, and food items were established based on consultation of key informants from the study area who were knowledgeable about the culture, and local language, and locally consumed foods. Various limitations to be considered when interpreting our results include the following: the cross-sectional nature of the data limits causal inference between the frequency of consumption of ASFs and their correspondences, and due to sample collection being from a single season, this limits the generalizability of the results to other reasons. In addition, due to individual differences in dietary consumption in the study setup over seven days, we establish our definition of the reference period of seven days. Women who could have eaten food items more than once in seven days were also tagged with those who consumed one time over seven days could underrate the amount used up is another limitation.

## Conclusion

We found low animal source food consumption among pregnant women in this predominantly rural setting. Women's educational level and milk cow ownership were positively

associated with ASF consumption. Additionally, a lower frequency of ASF consumption was observed among women who reported chewing khat. Raising women's awareness of the benefit of the intake of ASF in improving perinatal outcomes is suggested. Additionally, increased production of livestock at the household level and behavioral change communication on the intake of ASFs of women should promote shifts in social norms are essential. Promote shifts in social norms on the habit of chat chewing using religious leaders and influential community members to realize adequate nutrition for pregnant women social and behavioral change communication on maternal nutrition. Therefore, nutrition policy programs and interventions aimed at encouraging maternal nutritional guidance and counseling are recommended. Finally, we recommend the need for future further qualitative study using qualitative to explore barriers.

## Acknowledgments

Special thanks go to the Haramaya district health office staff for their enormous support during the data collection period. Finally, we like to thank all the women who participated in the study, the data collectors, and the supervisors.

## Author Contributions

**Conceptualization:** Abera Kenay Tura.

**Data curation:** Meseret Belete Fite, Abera Kenay Tura, Tesfaye Assebe Yadeta, Lemessa Oljira, Kedir Teji Roba.

**Formal analysis:** Meseret Belete Fite, Abera Kenay Tura, Tesfaye Assebe Yadeta, Lemessa Oljira, Kedir Teji Roba.

**Funding acquisition:** Meseret Belete Fite, Abera Kenay Tura, Tesfaye Assebe Yadeta, Lemessa Oljira.

**Investigation:** Meseret Belete Fite, Abera Kenay Tura, Tesfaye Assebe Yadeta, Lemessa Oljira, Kedir Teji Roba.

**Methodology:** Meseret Belete Fite, Abera Kenay Tura, Tesfaye Assebe Yadeta, Lemessa Oljira, Kedir Teji Roba.

**Project administration:** Meseret Belete Fite, Abera Kenay Tura, Tesfaye Assebe Yadeta, Lemessa Oljira, Kedir Teji Roba.

**Resources:** Meseret Belete Fite, Abera Kenay Tura, Tesfaye Assebe Yadeta, Lemessa Oljira, Kedir Teji Roba.

**Software:** Meseret Belete Fite, Abera Kenay Tura, Tesfaye Assebe Yadeta, Lemessa Oljira, Kedir Teji Roba.

**Supervision:** Meseret Belete Fite, Abera Kenay Tura, Tesfaye Assebe Yadeta, Lemessa Oljira, Kedir Teji Roba.

**Validation:** Meseret Belete Fite, Abera Kenay Tura, Tesfaye Assebe Yadeta, Lemessa Oljira, Kedir Teji Roba.

**Visualization:** Meseret Belete Fite, Abera Kenay Tura, Tesfaye Assebe Yadeta, Lemessa Oljira, Kedir Teji Roba.

**Writing – original draft:** Meseret Belete Fite, Abera Kenay Tura, Tesfaye Assebe Yadeta, Lemessa Oljira, Kedir Teji Roba.

**Writing – review & editing:** Meseret Belete Fite, Abera Kenay Tura, Tesfaye Assebe Yadeta.

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
