## [Decision Letter · Decision Letter 0]

14 Dec 2021

PONE-D-21-17077Dietary Practices and Associated Factors Among Pregnant Women in Haramaya District, Eastern EthiopiaPLOS ONE

Dear Dr. Fite,

Thank you for submitting your manuscript to PLOS ONE. After careful consideration, we feel that it has merit but does not fully meet PLOS ONE’s publication criteria as it currently stands. Therefore, we invite you to submit a revised version of the manuscript that addresses the points raised during the review process.

The manuscript has been evaluated by three reviewers, and their comments are available below.

The reviewers have raised a number of major concerns to the study methodology. They feel that considerations of potential confounding variables should be accounted for in the statistical technique used; and the introduction could be further expanded to describe how the current study further contributes to scientific knowledge in light of the articles which the reviewer has noted.

Finally, they feel that the copy editing is needed on the manuscript. Please note that one of the publication criteria at PLOS ONE (https://journals.plos.org/plosone/s/criteria-for-publication#loc-5 )is that articles must be presented in an intelligible fashion and written in clear, correct, and unambiguous English

Could you please carefully revise the manuscript to address all comments raised?

We look forward to receiving your revised manuscript.

Kind regards,

Lucinda Shen, MSc

Staff Editor

PLOS ONE

Journal Requirements:

2. Please include additional information regarding the survey or questionnaire used in the study and ensure that you have provided sufficient details that others could replicate the analyses. For instance, if you developed a questionnaire as part of this study and it is not under a copyright more restrictive than CC-BY, please include a copy, in both the original language and English, as Supporting Information. If the original language is written in non-Latin characters, for example Amharic, Chinese, or Korean, please use a file format that ensures these characters are visible.

“The authors would like express sincere appreciation to Haramaya University for funding of this study. Special thanks go to Haramaya district health office staffs for their enormous support in the whole period of data collection.  We acknowledge all study subjects volunteered for the study, data collectors and supervisors for their cooperation and meticulous contributions. “

“This study was fully funded by Haramaya University, Ethiopia. The funder has no role in conception, design of the study, statistical analysis, result interpretation and in writing up the manuscript. The funding institution has no role in the publication consent or approval.”

5. Please include a separate caption for each figure in your manuscript

Reviewers' comments:

Reviewer's Responses to Questions

**Comments to the Author**

1. Is the manuscript technically sound, and do the data support the conclusions?

Reviewer #1: Yes

Reviewer #2: Yes

Reviewer #3: No

2. Has the statistical analysis been performed appropriately and rigorously? 

Reviewer #1: No

Reviewer #2: Yes

Reviewer #3: Yes

3. Have the authors made all data underlying the findings in their manuscript fully available?

Reviewer #1: Yes

Reviewer #2: Yes

Reviewer #3: Yes

4. Is the manuscript presented in an intelligible fashion and written in standard English?

Reviewer #1: Yes

Reviewer #2: No

Reviewer #3: No

5. Review Comments to the Author

Reviewer #1: The regression model used by the authors does not seem appropriate. The outcome variable is binary.

The authors used Poisson regression which normally is used for count variables. The reason for the use of this model has not been clearly stated in the method section.

Reviewer #2: Dietary Practices and Associated Factors among Pregnant Women in Haramaya District, Eastern Ethiopia (PONE-D-21-17077).

PLOS ONE Comments to the Author

1. Is the manuscript technically sound, and do the data support the conclusions?

Yes.

2. Has the statistical analysis been performed appropriately and rigorously?

Yes

3. Have the authors made all data underlying the findings in their manuscript fully available?

Yes

4. Is the manuscript presented in an intelligible fashion and written in Standard English?

No

5. Review Comments to the Author

Summary

The manuscript is good. However, it does need an edition by a native English speaker or Journal English language services. Because PLOS ONE does not copyedit accepted manuscripts; the authors should employ an editor to assist with ambiguous and grammatical errors that appear throughout the text. There are multiple grammar and sentence structure corrections that are required prior to publication.

In title page And those authors listed above are also contributed equally to this work. Better if you use the symbol for the word and.

Please in include line number. Table 2: Factors associated with dietary practice among pregnant women in Haramaya district, Eastern Ethiopia, 2021. Table title should be written at the start.

Abstract

In your abstract part, please change the word Introduction by Background. in general, you have to follow the author guideline of the journal. In Method part Sectional study design what does it mean? Conclusion, in that dietary practice of pregnant women is sub-optimal. What is your reference to say sub-optimal.

Introduction

The introduction part is unclear. It should be written again coherently.

Method

Data collection and Measurement part is vague. Your operational definition is not clear. It should be rewritten again. Additionally, High DDS: Tercile calculated from food groups, and the highest tercile was considered as a high DDS, whereas the rest two lower terciles were taken as a low DDS.

High DDS: terciles were calculated from food groups, and the highest tercile was considered as a high DDS, whereas the rest two lower terciles were taken as low DDS (25). Please remove redundancy.

Factors Associated with Dietary Practice in this part you have written only the word table. But, you should include table number.

Again, Table 2: Factors associated with dietary practice among pregnant women in Haramaya district, Eastern Ethiopia,2021 is written at the end of your table which is not acceptable.

Table 2 in footnote you have incorporated definitions like Appropriate dietary practice: is defined as the consumption of at least four meals daily, high DDS, high FVS and high ASF; otherwise, Inappropriate dietary practice and Elementary school: Grade 1-8. But, it is redundant and better if you write it in your measurement part only.

Results

Table 1. Educational level of women is not standard classification. Additionally, in your footnote of your table you have included definitions like Housewife: Women whose activities are in home and not participated in yielding family financial income

Formal education+: Refers grade 1 and above

Wealth index quintile: Was computed from wealth score of the households and used to label households wealth status to five categories

Family Size: Refers to total number of family members living together

Parity: Refers to the number of births the mother experienced after 28 weeks whatever the status of the newborn is (WHO,2015). But, if you want to incorporate these definitions it is better to include in the measurement part. Finally, what is 1 mean?

Discussion

Adequate nutrition in pregnancy is important for maternal and child health (3). There is mounting evidences that, insufficient consumption of balanced and quality diet during pregnancy has a great effect on fetus health and development and may also result to poor birth outcome (4). At beginning of pregnancy many women lack sufficient micronutrients stores to meet physiological requirement (1) and they are more vulnerable to malnutrition (29). Several epidemiological studies indicated inappropriate dietary practice is significant contributor to maternal under nutrition and micronutrients deficiency in resource limited countries (30) .Your discussion part is very broad and somewhat not coherent. It needs a major revision.

Conclusion

Your conclusion is somewhat vague. Better if you write it clearly based on your finding.

References

Your reference is not written according to PLoS one in addition to some of them is very old. Please, review your references and adjust agreeing the PLOS specifications. For instance, you write some references in their complete journal’s title, whereas others were abbreviated.

6. PLOS authors have the option to publish the peer review history of their article (what does this mean?). If published, this will include your full peer review and any attached files.

Do you want your identity to be public for this peer review?

No

Reviewer #3: Dear Dr Meseret Belete

I am glad enough on your area of interest to do research on “Dietary Practices and Associated Factors among Pregnant Women in Haramaya District, Eastern Ethiopia” a means of prevention of feto-maternal adverse outcomes. Unfortunately, the whole of the manuscript lacks novelty and needs extensively edition due to a lot of typographical and grammatical errors which should be corrected. Otherwise unable proceed to further review. Again revise the word and line spacing across the manuscript. E.g., “asses” should be written as “assess”, “ meal frequency are measures usually” to be revised as “meal frequency measures are usually”, and lots punctuation errors.

Why you are intended to do this research due to many studies including meta-analysis on the area and what new evidence you provide for scientific community and nutrition experts? https://pubmed.ncbi.nlm.nih.gov/32855822 and https://pubmed.ncbi.nlm.nih.gov/34111140.

Chat chewing and restricted food intake is known factor for decrease diatery practice and might be having collinearly effect. How you control the effect of chat chewing on food restriction and dietary practice?

If you have the data better to do analysis by fitting other new variables in your regression. Hence, the variables you got are already explored by other studies. It was better to do mixed studies to explore qualitative barriers of adherence to recommendations of dietary diversity of pregnant women. Again, it needs project to alleviate the problem than repeated single research.

You suggest in the background “environmental and psychological factors in pregnancy are not well-known in this setup empirically”, but we can’t access it in the analysis.

How to measure dietary diversity among pregnant women, if you included those who were restricted of dietary/ food intake? Hence, you already put as a factor.

Please describe the possible determinants of dietary practice in Ethiopia and other settings?

Why intended to use Poisson regression analysis model?

Did you include women with emesis gravidarum before 12 weeks? And what specific illness was excluded in your actual data collection?

What does that mean 15.18% of pregnant women have dietary practice in your abstract section? appropriate dietary practice or HDD?

The write up of the results section needs extensive edition and less attractive the reader

Remove repeated words and used the journal guideline for the submission. E.g. put the table after the reference section with Table Legends, and double line spacing.

In discussion, you have expected to put your findings and compare with other studies and also suggest the possible implications of your study.

Regards

6. PLOS authors have the option to publish the peer review history of their article (what does this mean?). If published, this will include your full peer review and any attached files.

Reviewer #1: **Yes: **DEREJE TSEGAYE

Reviewer #2: No

Reviewer #3: **Yes: **melaku desta

---

## [Author Response · Author response to Decision Letter 0]

17 Jan 2022

I have tried to incorporate your comments

---

## [Decision Letter · Decision Letter 1]

20 May 2022

PONE-D-21-17077R1Consumption of animal source food and associated factors among pregnant women in Eastern Ethiopia: A community-based studyPLOS ONE

Dear Dr. Fite,

Thank you for submitting your manuscript to PLOS ONE. After careful consideration, we feel that it has merit but does not fully meet PLOS ONE’s publication criteria as it currently stands. Therefore, we invite you to submit a revised version of the manuscript that addresses the points raised during the review process.

We look forward to receiving your revised manuscript.

Kind regards,

Melaku Desta

Guest Editor

PLOS ONE

Journal Requirements:

Additional Editor Comments (if provided):

Dear authors of manuscript

It is highly worthful paper and an extensive revision was made based on the reviewers comment. Despite, to accept it for publication still needs further revision mainly on the typographical errors across the manuscript. Thus, lets correct the grammar errors.

Again, you have put the problem statement in clearly manner in the context of LMIC and Ethiopia instead of comparing with USA.

Reviewers' comments:

Reviewer's Responses to Questions

**Comments to the Author**

1. If the authors have adequately addressed your comments raised in a previous round of review and you feel that this manuscript is now acceptable for publication, you may indicate that here to bypass the “Comments to the Author” section, enter your conflict of interest statement in the “Confidential to Editor” section, and submit your "Accept" recommendation.

Reviewer #1: (No Response)

Reviewer #2: All comments have been addressed

2. Is the manuscript technically sound, and do the data support the conclusions?

Reviewer #1: Yes

Reviewer #2: Yes

3. Has the statistical analysis been performed appropriately and rigorously? 

Reviewer #1: Yes

Reviewer #2: Yes

4. Have the authors made all data underlying the findings in their manuscript fully available?

Reviewer #1: Yes

Reviewer #2: Yes

5. Is the manuscript presented in an intelligible fashion and written in standard English?

Reviewer #1: No

Reviewer #2: Yes

6. Review Comments to the Author

Reviewer #1: Th authors have tried to address the comments. However, the manuscript has a lot of grammatical errors. Needs proof reading.

Reviewer #2: Authors have made a significant improvement in their revised document. It now looks amended and clear to the reader.

7. PLOS authors have the option to publish the peer review history of their article (what does this mean?). If published, this will include your full peer review and any attached files.

Reviewer #1: **Yes: **Dereje Tsegaye

Reviewer #2: No

---

## [Editor Report · Decision Letter 2]

31 May 2022

PONE-D-21-17077R2Consumption of animal source food and associated factors among pregnant women in Eastern Ethiopia: A community-based studyPLOS ONE

Dear Dr. Fite,

Thank you for submitting your manuscript to PLOS ONE. After careful consideration, we feel that it has merit but does not fully meet PLOS ONE’s publication criteria as it currently stands. Therefore, we invite you to submit a revised version of the manuscript that addresses the points raised during the review process.

We look forward to receiving your revised manuscript.

Kind regards,

Melaku Desta

Guest Editor

PLOS ONE
---

## [Editor Report · Decision Letter 3]

8 Jun 2022

Consumption of animal source food and associated factors among pregnant women in eastern Ethiopia: A community-based study

PONE-D-21-17077R3

Dear Dr. Fite

We’re pleased to inform you that your manuscript has been judged scientifically suitable for publication and will be formally accepted for publication once it meets all outstanding technical requirements.

Kind regards,

Melaku Desta

Guest Editor

PLOS ONE
---

## [Editor Report · Acceptance letter]

10 Jun 2022

PONE-D-21-17077R3 

Consumption of animal source food and associated factors among pregnant women in eastern Ethiopia: A community-based study 

Dear Dr. Fite:

I'm pleased to inform you that your manuscript has been deemed suitable for publication in PLOS ONE. Congratulations! Your manuscript is now with our production department. 

Kind regards, 

on behalf of

Dr. Melaku Desta 

Guest Editor

PLOS ONE